# Percutaneous CT-Guided Bone Biopsies: Indications, Feasibility and Diagnostic Yield in the Different Skeletal Sites—From the Skull to the Toe

**DOI:** 10.3390/diagnostics13142350

**Published:** 2023-07-12

**Authors:** Paolo Spinnato, Marco Colangeli, Raffaella Rinaldi, Federico Ponti

**Affiliations:** 1Diagnostic and Interventional Radiology, IRCCS Istituto Ortopedico Rizzoli, 40136 Bologna, Italy; 2Orthopaedic Oncology Unit, IRCCS Istituto Ortopedico Rizzoli, 40136 Bologna, Italy

**Keywords:** bone and bones, bone neoplasm, image-guided biopsy, multidetector computed tomography, neoplasm metastasis, sarcoma

## Abstract

CT-guided bone biopsies are currently the diagnostic tool of choice for histopathological (and microbiological) diagnoses of skeletal lesions. Several research works have well-demonstrated their safety and feasibility in almost all skeletal regions. This comprehensive review article aims at summarizing the general concepts in regard to bone biopsy procedures, current clinical indications, the feasibility and the diagnostic yield in different skeletal sites, particularly in the most delicate and difficult-to-reach ones. The choice of the correct imaging guidance and factors affecting the diagnostic rate, as well as possible complications, will also be discussed. Since the diagnostic yield, technical difficulties, and complications risk of a CT-guided bone biopsy significantly vary depending on the different skeletal sites, subdivided analyses of different anatomical sites are provided. The information included in the current review article may be useful for clinicians assisting patients with possible bone neoplasms, as well as radiologists involved in the imaging diagnoses of skeletal lesions and/or in performing bone biopsies.

## 1. Introduction

Biopsies can be considered as the ultimate diagnostic tool in the diagnostic flowchart of skeletal lesions. As the ultimate step, it should always be preceded by an accurate clinical and radiological assessment.

The goal of the biopsy procedure is to obtain a tissue sample that permits to obtain a diagnosis with the minimum risk to the patients. Due to this, minimally invasive approaches such as percutaneous needle biopsies are currently preferred to surgical ones as the first line. Image-guided biopsy procedures have been performed increasingly frequently in recent decades, due to the reduction in complications and increased precision compared to blind procedures.

The goal of this comprehensive review article is to offer a wide overview on this topic, useful for clinicians involved in the care of patients with bone neoplasm, offering an update on the indications, potentialities and limitations of this tool. Nonetheless, this article aims to serve as a practical guide for radiologists performing bone biopsies and/or involved in bone tumor imaging diagnosis.

### 1.1. Biopsy: General Concepts

A biopsy can be performed surgically (open biopsy) or by a percutaneous approach (closed biopsy). An open biopsy presents several disadvantages, including increased procedure times, risks inherent to the surgical procedure (e.g., wound healing and infections), increased costs and the need for a hospital stay and deep anesthesia [1].

Skeletal biopsies are nowadays performed with the percutaneous approach, while open biopsies are generally performed after a non-diagnostic closed biopsy, especially in the extremities [2]. A percutaneous skeletal biopsy includes fine needle aspiration and also a core needle biopsy.

Imaging guidance offers additional value to the biopsy procedure in terms of the efficacy and safety compared to blind bone samples in all cases.

Fluoroscopy has been used as the main imaging tool for skeletal biopsy guidance for decades. Subsequently, with the introduction of cross-sectional imaging, computed tomography (CT) has been proposed as a guidance for bone biopsies. Nowadays, CT-guided biopsies are the preferred technique for all skeletal sites with increased advantages compared to fluoroscopy, particularly in the spine and pelvis [2]. In all skeletal regions, the precision and safety provided by CT are significant benefits.

### 1.2. History

In 1930, Martin and Ellis published a presentation of technical percutaneous procedures used to secure tissue from suspected neoplasms for a histological evaluation by needle puncture and aspiration [3]. This preliminary series included different body districts sampled, including several skeletal regions (the mandible, humerus, scapula, humerus and pelvis). The percutaneous procedures performed on bone resulted in successful histological diagnoses including bone sarcomas, chondrosarcomas, spindle cell sarcoma and osteogenic sarcoma [3].

One year later, Bradley and colleagues reported 35 cases of bone tumors successfully diagnosed by percutaneous needle aspiration [4]. These procedures, performed almost 100 years ago, represent the progenitor of modern skeletal biopsies, and in some way, of the entirety of percutaneous musculoskeletal interventional radiology. In the following decades, bone biopsies were associated with imaging guidance, firstly fluoroscopy, to increase precision and safety. Later on, together with the increased availability of cross-sectional imaging tools, skeletal biopsies were increasingly associated with CT guidance, gaining additional potentiality and nonetheless being recognized as a procedure for medical radiologists [5].

### 1.3. Image Guidance

A skeletal biopsy can be performed under fluoroscopic, CT or ultrasound guidance. CT guidance is currently the most used tool for the imaging of bone biopsies, permitting to sample almost all skeletal sites in a safe manner.

Fluoroscopy-guided skeletal biopsies still have a role, usually where CT guidance is not available. This procedure is generally performed in surgical settings by orthopedics. This procedure is particularly indicated for long bones in the extremities, where the precision of CT guidance is not strictly necessary [2].

An ultrasound-guided skeletal biopsy is a very interesting tool, offering the advantages of a real-time evaluation and an accurate adjacent soft tissue assessment, with particular regard to neurovascular bundles [6]. Nonetheless, this procedure is particularly indicated when a cortical disruption is present, aiding the ultrasound waves to bone lesions/neoplasms, and/or in the case of superficial bone regions such as the ribs [7,8]. In these cases, the diagnostic accuracy and safety are reported to be comparable with CT-guided biopsies.

The feasible and suggested imaging guidance techniques depending on the skeletal site and the lesions’ radiologic patterns are summarized in Table 1.

Additionally, several studies have underlined the feasibility, safety and efficacy of MRI guidance for bone biopsies. The advantages of MR imaging guidance are several. MRI uses no ionizing radiation in contrast to fluoroscopy and CT, provides improved bone marrow and soft tissue assessment and has multiplanar capabilities. Additionally, some bone lesions may only be visible on MRI. Current routine applications of this tool in clinical settings are still limited, mainly because of the long acquisition time and high costs.

In a large series by Liu et al. including 67 MRI-guided spinal biopsies, safe and accurate diagnostic results were reported, but a relatively high procedural time from 27 to 56 min (mean 35 min) [9].

### 1.4. Clinical Indications

Nowadays, CT-guided skeletal biopsies are considered the suggested first line tool for histopathologic and/or microbiologic diagnoses of bone lesions. This tool is the method of choice for the evaluation of skeletal lesions suspected to be malignant, i.e., suspicious primary bone tumors or systemic cancer metastasis.

The indications for imaging-guided skeletal biopsy are several. One of the most common indications is confirming metastasis in a patient with a known primary malignancy (Figure 1).

Even if the radiological studies together with anamnestic data indicate a diagnosis of bone metastasis, a CT-guided biopsy is indicated not only for histopathologic confirmation. Indeed, by deepening metastatic bone lesion profiling, molecular and receptor characterization can be obtained, with relevant impacts on the patient management and treatment choice [9,10]. Particularly, the hormone receptor status can be safely and accurately assessed on bone metastases with CT-guided samples, permitting the possibility of optimal treatments and a more personalized care [10].

Recurrence identification and tumor treatments response (e.g., % of necrosis obtained after therapy) are other well-known indications for bone biopsies [11]. Moreover, a bone biopsy is indicated when an unexpected non-traumatic fracture occurs. Particularly, determining the nature of a non-traumatic vertebral collapse is a recognized indication for these procedures [12].

Investigating for infection (microbiological analyses) or the differential diagnosis between infections (particularly spondylodiscitis) and other bone lesions is another indication as well [13,14,15,16]. Spondylodiscitis is usually diagnosed by a combination of clinical, laboratorial and imaging data. A percutaneous biopsy is usually recommended to isolate the pathogenic microorganisms in patients with negative blood cultures. Despite its low microbiologic yield, a CT-guided percutaneous biopsy is frequently preferred over an open biopsy due to its relative safety, low morbidity and inexpensive cost. Moreover, deep soft tissue infections, in selected cases and especially if bone involvement is present, may require a CT-guided biopsy.

It is important to know that in the case of primary bone sarcomas of the appendicular skeleton, a surgical incisional biopsy permits to obtain larger samples with the aim of deepening analyses for molecular and therapeutic investigations [17]. Due to this, when a radiographic feature of primary bone sarcoma is clearly recognized, the patient could be submitted directly for an incisional surgical biopsy.

Nonetheless, CT-guided biopsies may be preferred to the standard ultrasound guidance for selected deep soft tissue neoplasms, especially if bone erosion is present [18,19].

In Figure 2, we summarize all the main clinical indications for a bone biopsy.

### 1.5. Contraindications and Complications

The main contraindications are represented by coagulation disorders and anticoagulant treatments, increasing the risk of local bleeding/hematoma, and a decreased platelet count (<50,000/mm^3^). The Society of Interventional Radiology recommends correction of an INR (international normalized ratio) to less than 1.5–1.8. If the platelet count is fewer than 50,000/mm^3^, platelet transfusion may be necessary and anticoagulant and antiplatelet medications may be stopped depending on the unique pharmacologic properties of each drug [20].

Complications are reported to be very infrequent and usually mild. The main complications are transient paresis (due to the anesthetic effect on nerves) (0.09%) and hematoma (0.002%) [21]. Transient and minor complications such as vasovagal syncope are commonly observed, especially in the supine position rather than the prone position.

Pain during the procedure is a relevant concern, since bone lesions and the biopsy itself may be extremely painful in some cases. Local anesthetic, deep/mild sedation or general anesthesia and neural blocks should be carefully planned according to the lesion type and location and the patient’s age.

## 2. CT-Guided Skeletal Biopsy: Feasibility and Diagnostic Yield in Different Skeletal Sites

Almost all skeletal regions from the skull to the toes are safely accessible with CT guidance. The complexity of the procedure varies consistently according to different regions.

In the largest series focused on CT-guided bone biopsies performed in miscellaneous skeletal sites, the diagnostic yield was assessed in approximately 90% of cases. Rimondi et al., in a series of 2027 CT-guided bone biopsies, reported a diagnostic accuracy of 94%, including diagnoses obtained with biopsy repetition in 408 patients [21].

In the following sub-sections, we will explain in depth the feasibility and results of CT-guided biopsies applied in the most difficult or delicate skeletal sites: the skull, the craniovertebral junction and atlantoaxial spine (C0–C3), the spine, the ribs, fingers and toes.

### 2.1. Skull

There are very few reports in the current literature in regard to CT-guided skull biopsies. Indeed, in most clinical settings, biopsies of the skull are performed by neurosurgeons as open surgical procedures.

Despite this, research focused on CT-guided skull biopsies is very promising, underlining the feasibility, safety and efficacy of this diagnostic tool applied even in this uncommon skeletal site (Figure 3).

Tomasian et al. in 2019 reported the successful and safe application of CT-guided biopsies in 14 patients with skull lesions. In 13 out of 14 patients, a conscious sedation was applied, and a histologic diagnosis was obtained in 12 cases (86%) without complications recorded during or after the procedures [22].

Sundararajan et al. in 2022 reported a similar successful series of 12 low-dose CT-guided biopsies performed on calvarial bone lesions [23]. These preliminary available data suggest that the application of CT-guided biopsies in skull lesions should be considered in clinical practice to avoid a surgical approach. Larger series confirming the safety and efficacy of this tool in these sites are necessary.

### 2.2. Skull Base and Craniovertebral Junction (C0–C2)

Central skull base lesions are difficult to approach even with the aid of CT. Despite this, several successful strategies include the CT gantry obliquity ‘tilt angle’ and peculiar anatomical approaches [24,25]. For bone lesions located in the sphenoid or clivus/skull base bones, an endoscopic trans-sphenoidal biopsy can be considered as an alternative to craniotomy and CT-guided biopsies [26].

Non-invasive image processing tools are being developed to differentiate the most common primary tumor of this skeletal tract (chondrosarcoma and chordoma among primary tumors of bone) [27]; however, their application in clinical practice is still not feasible.

The craniovertebral junction (C0–C2), defined as a complex bony region composed of the occiput, the axis (C1) and the atlas (C2), is probably the most delicate skeletal site of the human skeleton, due to the presence of a vital neurovascular structure. CT guidance, with the aid of intravenous contrast media injection for the visualization of vertebral arteries if necessary, has been proven to be feasible and safe even in this complex skeletal region (Figure 4).

Spinnato et al. in 2022 reported the feasibility and safety of CT-guided bone biopsies performed in the craniovertebral junction of 16 patients, with a diagnostic accuracy of 81.3% [28]. In this series, the needle approach was always posterior or lateral, but it is well known that in peculiar atlantoaxial spinal locations (e.g., dens of the atlas), an anterior open-mouth approach through the oropharyngeal space should be chosen [29,30]. Using a contrast media injection can help visualize vertebral arteries, aiding a safe posterior (or lateral) approach [28].

### 2.3. The Spine

CT is considered the best modality to guide biopsies within the spine in all tracts. Several approaches can be taken depending on the lesion site and features. The trans-pedicle approach is the safest and is used to reach the vertebral body (Figure 5).

A thoracic spine costovertebral approach can also be considered [31]. Many other approaches can be performed in other less common cases, including the above-mentioned anterior approach in the atlanto-axial spine [29,30,31].

Rimondi et al., in a series of 430 CT-guided spinal biopsies, reported a diagnostic accuracy of 93.3%, and only 0.2% minor complications (five transient paresis and four spontaneously resolved hematomas) [32]. Similar results have been reported in research by Lis et al. analyzing the performance in 410 consecutive CT-guided biopsies of the spine with a diagnostic rate of 89% and 11% false negative results [33]. Many other several large series have confirmed the safety of this procedure for spinal lesion diagnosis [34].

Since nowadays CT guidance is by far preferred over fluoroscopic guidance for sampling most spinal locations, even if the first procedure results in a non-diagnostic result, repetition of a CT-guided biopsy is suggested before any other biopsy technique [2].

### 2.4. Ribs and Sternum

Lesions located in the rib cage and sternum represent a challenging site for bone biopsies in regard to the proximity of neurovascular bundles, the mediastinum, the pleura and the lungs, as well as abdominal organs. CT represents a safe guidance for rib biopsies, as well as ultrasound in selected cases (see Table 1). Many malignant (metastases and myeloma) and benign lesions may affect the ribs (fibrous dysplasia above all) and sternum (Figure 6).

A large series by Baffour et al. reported a diagnostic rate of 96.8% in 249 rib biopsies performed under CT guidance; 5.6% minor complications and no major complications were reported [35]. In this series, about 70% of diagnosis were malignant while the remainder were benign bone tumors.

A series by Song et al. including 34 sternal biopsies revealed a lower diagnostic rate (64.7%) compared to other skeletal sites. A relevant percentage of non-diagnosed cases (23.5%) were finally clinically diagnosed with inflammatory arthritis [36].

### 2.5. The Pelvis

The difficulty of biopsies performed in the bones of the pelvis is variable, depending on the site, the patient’s morphology (e.g., soft tissue thickness) and the aim of the procedure. Indeed, biopsies of the pelvic bones can be performed to target a focal lesion or to sample the bone marrow [37,38,39]. In the latter case, a bone marrow biopsy is usually performed in the staging of several malignancies and on the iliac crest without any imaging guidance. The addition of CT guidance even for bone marrow samples was shown to be superior to blinded biopsies, offering several advantages, in particular more adequate samples [40].

When a target lesion is present, the advantages of CT guidance are more evident. For example, for spinal locations and in pelvic bone locations, repetition of a CT-guided biopsy is suggested as the tool of choice even after a first non-diagnostic procedure [2].

### 2.6. Appendicular Skeleton

The appendicular skeleton can be a challenging site for bone biopsies. A lower diagnostic rate is expected if the lesion is located in the long bones. This is mainly due to the lower content of medullary bones in these bony segments, particularly in long bone diaphysis.

#### Hands and Feet

To the best of our knowledge, there are no series/original articles focused on CT-guided biopsies of the hands and feet. The procedure is known to be feasible in these skeletal regions from the results of research articles including all skeletal sites. Indeed, in several series of CT-guided biopsies performed in miscellaneous skeletal tracts, the hands and feet are included. In a large series of 2027 biopsies by Rimondi et al., only 25 (1.2%) were performed in the hands, and 5 in the feet (0.2%) [21]. This underlines that the procedure is feasible but is performed in these skeletal regions very rarely in clinical practice (Figure 7).

The peculiar anatomy and closeness to numerous small neurovascular structures suggest that future series focused on these skeletal regions, particularly on fingers and toes, should be performed to confirm the safety and effectiveness in these locations.

## 3. Factors Influencing the Diagnostic Yield

There are several well-recognized factors that can influence the results of CT-guided biopsies in terms of the diagnostic yield. Above all, the size of the lesion is a key element. Indeed, larger lesions (>3 cm) present a higher diagnostic rate compared to smaller ones [40]. Interestingly, the radiological pattern of bone lesions may also influence the diagnostic accuracy. Sclerotic/osteoblastic lesions are reported to present a lower diagnostic rate compared to osteolytic or mixed ones [40,41,42]. To this aim, recent research by Donners et al. focused on cancer patients with sclerotic bone disease, revealing that sampling areas of predominantly mild sclerosis rather than severe sclerosis can improve the tumor tissue yield and biopsy feasibility [36]. In detail, moderate sclerotic lesions (mean Hounsfield unit = 244) resulted in a diagnostic specimen in 87% of cases. On the contrary, in dense sclerosis (mean Hounsfield unit = 622), the diagnostic yield was 56% (Figure 8) [41].

Wu et al., in a series of 151 image-guided bone biopsies, found that the diagnostic yield was significantly different depending on the radiological pattern of the lesion (*p* = 0.002); it was 87% for lytic lesions and 57% for sclerotic ones [43]. Moreover, larger lesions are known to be associated with a higher rate of diagnostic success compared to smaller ones. The number of specimens obtained influenced the diagnostic yield as well. Indeed, obtaining more than one sample (ideally three) is suggested [43]. Moreover, the location of the lesions in the long bones of the appendicular skeleton is another factor determining a lower diagnostic rate compared to other skeletal sites [43]. Another important factor affecting the diagnostic yield is the longer penetration distance of the tumor by the biopsy needle, associated with a higher diagnostic yield. Indeed, longer specimens are associated with a higher successful diagnostic rate [43]. To obtain a higher diagnostic rate in small lesions or in small bones, an oblique penetration (avoiding perpendicular needle angulation) is suggested. In suspected spondylodiscitis, targeting the paravertebral soft tissue involved or fluid collection rather than bone could increase the microbiologic diagnostic yield [44]. More generally, in osteomyelitis, the aspiration of fluid collections with at least 2 mL of purulent fluid is associated with a higher microbiological diagnostic yield [45]. Importantly, the microbiological diagnostic yield does not increase in the case of antibiotic use (prior to biopsy) [46].

A recent series of 1033 CT-guided bone biopsies found out that when the target lesion is not visible on CT scans the diagnostic yield was significantly lower compared to lesions visible on CT, at 37.1% and 76.9%, respectively (*p* < 0.001) [47]. In the case of a non-visible lesion on CT, a biopsy was performed in the skeletal sites involved as documented on MRI and/or PET/CT. Hematological malignancies, particularly leukemia and lymphoma, as well as red marrow, were more common among occult lesions on CT compared to visible ones [47].

Finally, it is important and interesting to know that the final histopathological diagnosis influences the diagnostic yield as well. Indeed, benign final diagnoses are associated with a lower diagnostic yield (and frequent biopsy repetition) compared to malignant ones [36,48]. An exception can be considered for malignant neoplasms with a high necrotic component. In these cases, the biopsy can result in non-diagnostic results if only the necrotic tissue is sampled [49]. Since necrosis usually localizes in central areas, sampling the tumor borders and carefully evaluating pre-operative imaging studies (contrast enhanced CT, MRI or PET-CT scans) can help in the reduction in non-diagnostic necrotic samples.

All the main factors affecting the diagnostic yield of CT-guided skeletal biopsies are summarized in Table 2.

In non-diagnostic CT-guided biopsies, according to the clinical setting and multidisciplinary evaluations, the procedure can be repeated under CT-guidance again for spinal, pelvic and skull locations. On the contrary, for appendicular skeletal lesions, repetition of the biopsy can be performed with incisional surgery to obtain larger samples [2].

## 4. Conclusions

An accurate biopsy is essential in the diagnosis of bone lesions. The safety and effectiveness of CT-guided biopsies applied in almost all skeletal sites render this tool a key element in the diagnostic work-up and management of patients with suspected or known bone tumors, as well as of patients affected by systemic malignancies. Indeed, the indications for bone biopsies have increased over the last decades. A more personalized tissue characterization provides a deeper molecular analysis of a suspected or known malignancy, aiding the most appropriate care.

In the future, technological advancements applied to imaging may help to optimize image-guided biopsies, increasing the accuracy of this tool, for example, the use of fusion imaging techniques allowing the superimposition of CT or US images with MRI and/or PET studies [50].

Oncologists, orthopedic surgeons and radiologists, as well as other physicians involved in patient care, should know the potential, indications and limits of this tool. Moreover, it is important to know the differences in the feasibility, difficulty and diagnostic rate of image-guided biopsies in different skeletal sites.

## Figures and Tables

**Figure 1 diagnostics-13-02350-f001:**
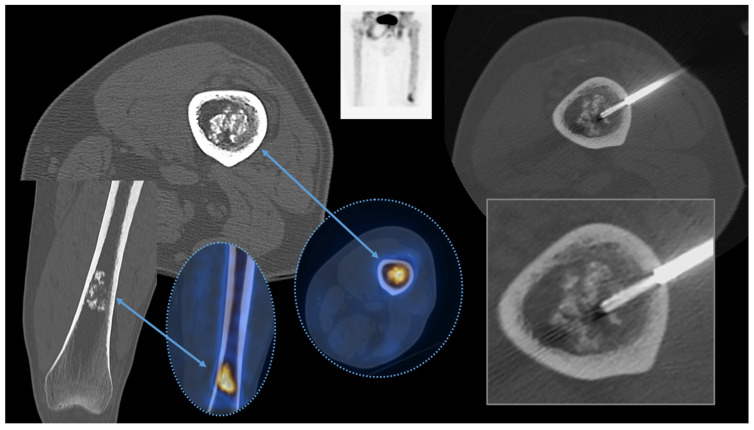
A male patient with a recent diagnosis of prostate cancer was put forward for a CT-guided biopsy (right, square) of a calcified intramedullary lesion of the distal femur metaphysis with increased SUV at PET-CT (oval dotted lines). A diagnosis of enchondroma, already suspected by radiological reports, was confirmed by a biopsy, excluding metastasis.

**Figure 2 diagnostics-13-02350-f002:**
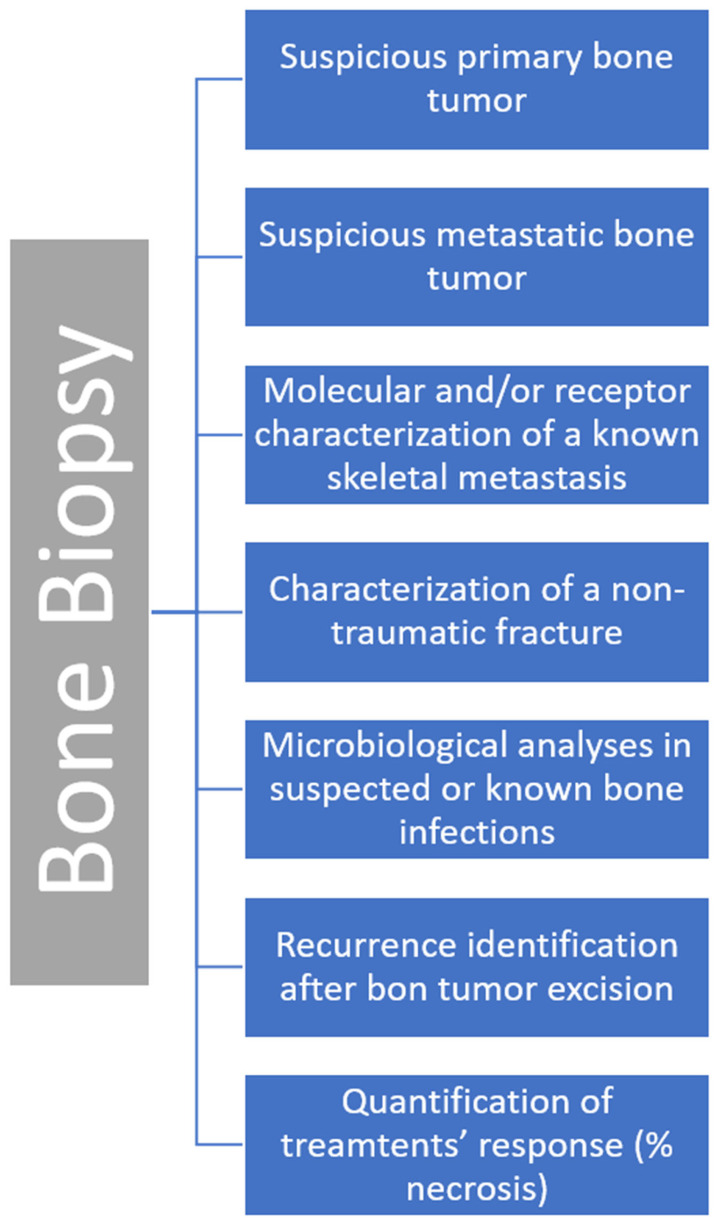
Summary of clinical indications for a bone biopsy.

**Figure 3 diagnostics-13-02350-f003:**
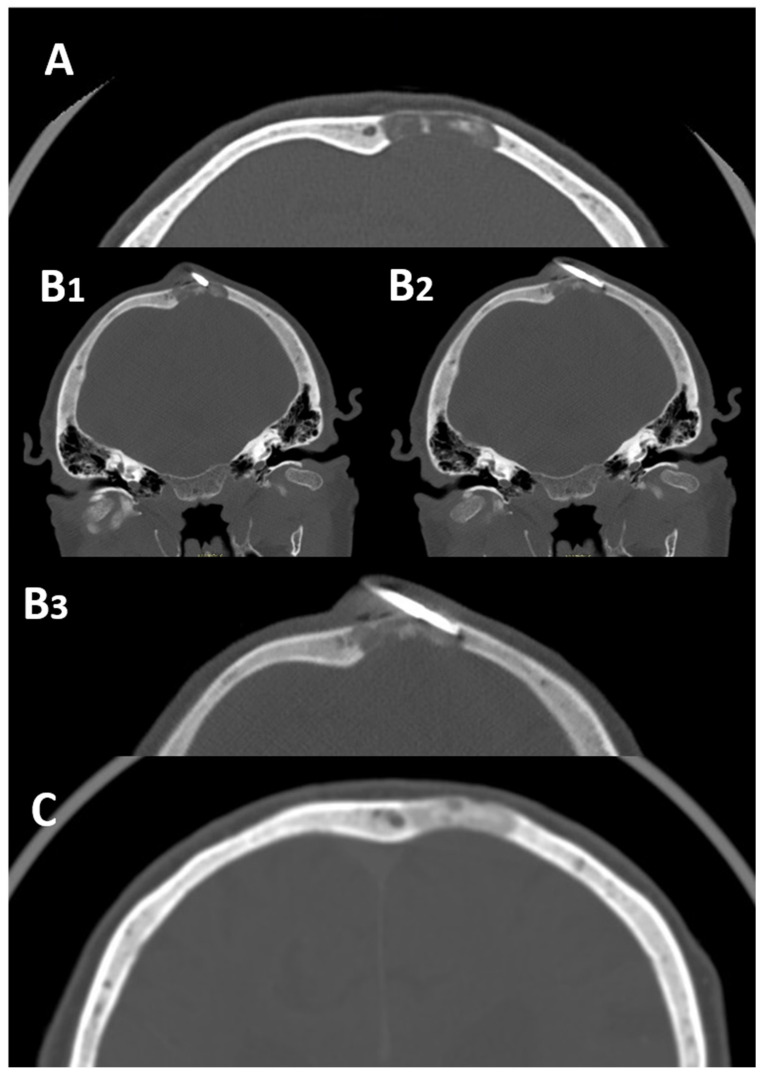
85-year-old female with a solitary osteolytic bone lesion of the occiput (axial CT panel (**A**)) and previous history of breast cancer. The patient was submitted to a CT-guided biopsy to avoid possible complications of surgery and anesthesia, in regard to the patient’s age. The insertion of the needle inside the lesion was checked with CT guidance with technical success achieved (Panel (**B1**–**B3**) enlargement). The biopsy was diagnostic and a bone metastasis from breast cancer was confirmed. The patient was subsequently submitted to radiation therapy with the optimal response and complete calcification of the lesions (Panel (**C**)).

**Figure 4 diagnostics-13-02350-f004:**
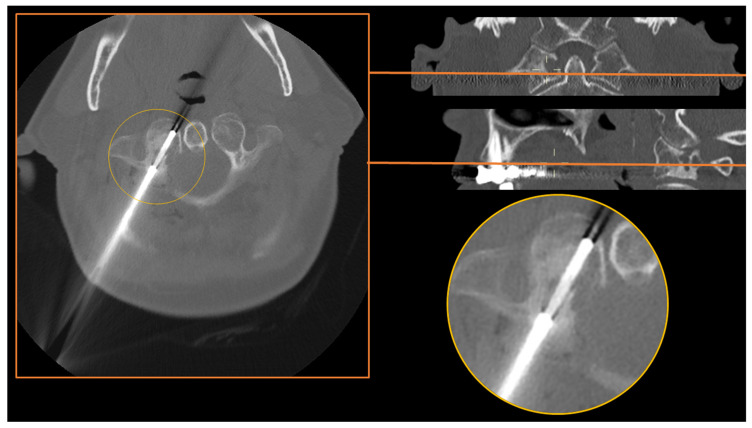
A CT-guided biopsy with a posterior approach (round circles) of the right lateral mass of C1 (atlas) in a woman with a mixed lesion incidentally diagnosed in a maxillofacial CT. The final diagnosis was osteomyelitis, and malignancies have been excluded.

**Figure 5 diagnostics-13-02350-f005:**
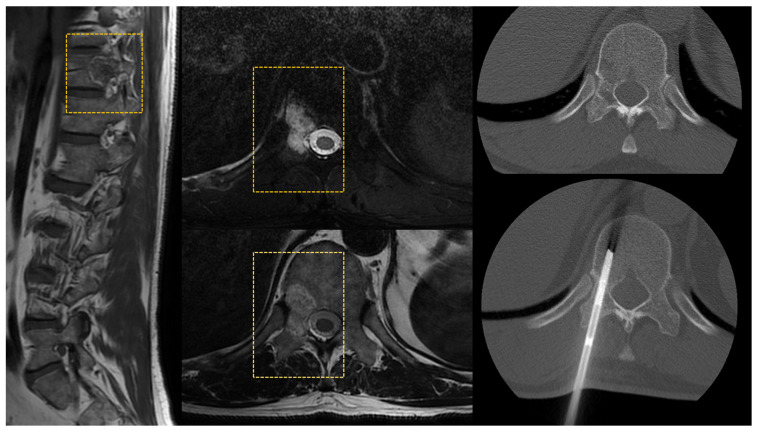
Vertebral lesion of the right pedicle and body of T11 diagnosed with MRI (square dotted lines). A CT-guided right trans-pedicle needle biopsy confirmed the diagnosis of hemangioma.

**Figure 6 diagnostics-13-02350-f006:**
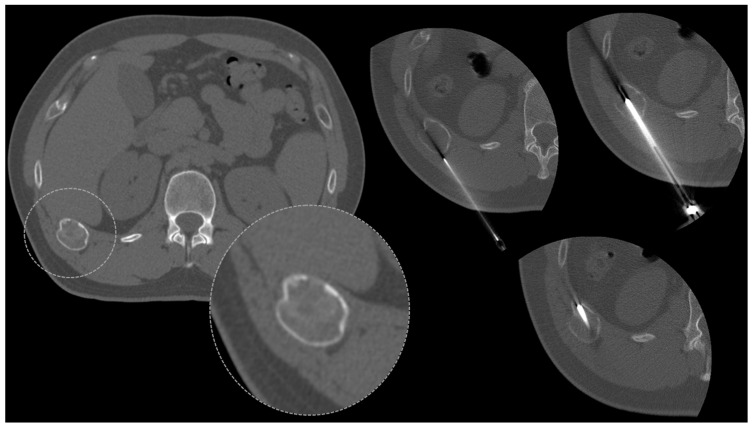
Osteolytic rib lesion with osseous enlargement and cortical thinning (oval dotted lines), biopsied under CT-guidance (right).

**Figure 7 diagnostics-13-02350-f007:**
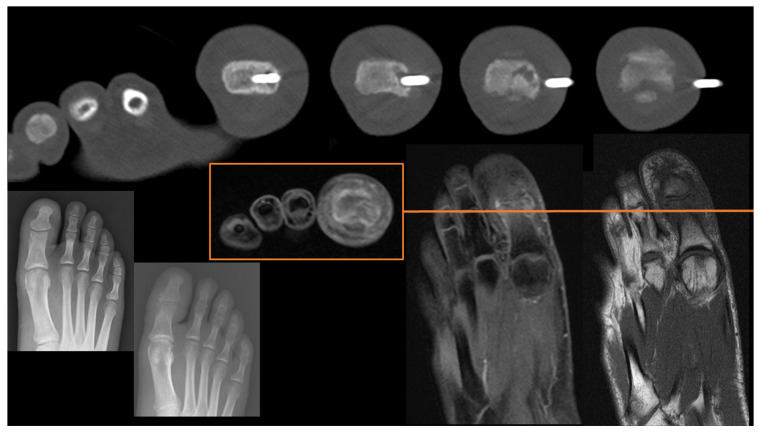
Osteolytic bone lesion located in the hallux: conventional radiography (bottom left), MRI (bottom right and orange square) and CT-guided needle biopsy (up). The final diagnosis obtained was osteomyelitis.

**Figure 8 diagnostics-13-02350-f008:**
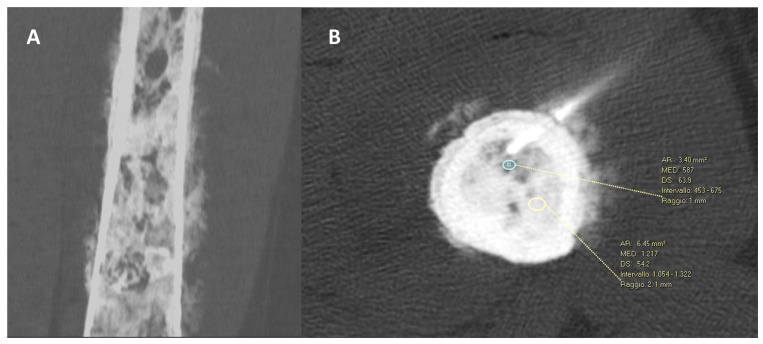
(**A**) CT scan, coronal reconstruction, in a 43-year-old woman showing a diffuse sclerotic lesion of the femoral diaphysis with an aggressive cortical reaction pattern. (**B**) A CT-guided biopsy was performed in the less dense intramedullary component of the tumor (blue oval line—587 HU) avoiding the more dense component (yellow oval line—1217 HU). The CT-guided biopsy resulted in a successful diagnostic sample with a final diagnosis of high-grade osteoblastic osteosarcoma.

**Table 1 diagnostics-13-02350-t001:** Summary of imaging guidance possibilities for bone biopsies in different skeletal sites according to lesion radiologic patterns.

Affected Skeletal Region	Radiologic Pattern	Imaging Guidance
Appendicular skeleton	Sclerotic pattern, osteolytic or mixed pattern without cortical disruption or intramedullary only involvement	**CT**, Fluoroscopy
Appendicular Skeleton	Osteolytic or mixed pattern with cortical disruption or cortical thinning	Ultrasound, **CT**, Fluoroscopy
Pelvic bones	Lesions without large extra-skeletal soft tissue involvement	**CT**
Pelvic bones	Osteolytic or mixed with large soft tissue extra-skeletal involvement	**CT**, Ultrasound
Spine	All	**CT**
Spine	Large lesions of the vertebral body	**CT**, Fluoroscopy
Spine	Large lesions with posterior extra-osseous involvement	**CT**, Ultrasound
Ribs	Osteolytic or mixed pattern with cortical disruption or cortical thinning	**CT**, Ultrasound
Ribs	Sclerotic pattern, osteolytic or mixed pattern without cortical disruption or intramedullary only involvement	**CT**
Sternum	All	**CT**
Sternum	Anterior extra-skeletal soft tissue involvement	**CT**, Ultrasound
Skull and Craniovertebral junction	All	**CT**

^1^ CT = computed tomography.

**Table 2 diagnostics-13-02350-t002:** Summary of factors associated with lower and higher diagnostic rates.

Factors Associated with a Lower Diagnostic Rate	Factors Associated with a Higher Diagnostic Rate
Sclerotic radiologic pattern	Lytic or mixed radiologic pattern
Location in the appendicular skeleton	Location in the axial skeleton
Small lesion size	Large lesion size
Single bone sample	Multiple bone samples
Short sample (short needle penetration in the tumor/perpendicular needle trajectory)	Large sample (long needle penetration in the tumor/oblique needle trajectory)
(Spondylodiscitis/Osteomyelitis) Targeting the bone only	(Spondylodiscitis/Osteomyelitis) Targeting soft tissue involvement and/or fluid collection aspiration.
Target lesion not visible in CT	Target lesion visible in CT
Benign final diagnosis	Malignant final diagnosis

## Data Availability

No data available for this review article.

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
