# Peer review of "Percutaneous CT-Guided Bone Biopsies: Indications, Feasibility and Diagnostic Yield in the Different Skeletal Sites—From the Skull to the Toe"

_diagnostics, 2023, doi:10.3390/diagnostics13142350_

Round 1
Reviewer 1 Report
Firstly, I would like to thank you for inviting me to review the manuscript entitled “Percutaneous CT-guided Bone Biopsy: Indications, Feasibility and Diagnostic Yield in Different Skeletal Sites - From the Skull to the Toe.”
The authors present a comprehensive review of percutaneous CT-guided bone biopsies. The title accurately reflects the case. The manuscript is well written in terms of clarity, style, and use of English and has a logical construction. The conclusions accurately explain the main clinical message. The figures are of good quality and relevant to the clinical message. The references are appropriate and current.
I can detect no significant flaws in the manuscript.
Minor comment:
In the discussion of factors affecting the diagnostic yield of CT-guided skeletal biopsy, the authors should include that in cases of malignant neoplasms with extensive necrosis, the biopsy tissue could consist of non-viable tissue.
Author Response
Thank You very much for the great appreciation of our manuscript!
As requested we added intratumoral necrosis as a possible cause of non diagnostic biopsy specimen.
Thank you.
Reviewer 2 Report
It's an interesting paper with a large number of explanation of cases and potentiality of this technique
Author Response
Thank You very much for the high appreciation of our paper!
We are honored.
Thanks
Reviewer 3 Report
CT biopsy, as an important diagnostic tool for the histopathological diagnosis of bone lesions, has been widely used in the detection of bone lesions in various parts of the human body, and has made contact contributions. On the basis of existing literature, this paper summarizes and further analyzes and discusses the origin, application scope and factors affecting the diagnostic delivery rate of CT biopsy. Since most of the diseases are special, this paper also summarizes many personalized cases, which is of great reference value for clinical doctors and other technical personnel. I think this article can be modified and published in this Journal. The comments are as follows:
1.Please note the uniform format of the content of the article, especially the indentation before the paragraph. In addition, please pay attention to the logical words and grammar problems in the structure of the essay.
2.Article title missing: Section 2.1 title missing before section 2.1.1 and Section 2.1.2. Also, without chapter 3, skip straight to chapter 4.
3.The picture format needs to be changed. First of all, whether the picture is original by the author, if not, please indicate the source. In addition, the title of the picture should be summarized in a short sentence, and the detailed description of the picture should be moved to the body text. Finally, please mark some of the main features in the picture. If there are multiple pictures in one picture, please mark them separately.
4.In the introduction part, some supplementary literatures should be added for the extensive application of a certain technology and some conclusive expressions, and a general conclusion cannot be drawn from just one or two literatures. For example, " Skeletal biopsies are nowadays performed with the percutaneous approach, while open biopsies are generally performed after a non-diagnostic closed biopsy, especially in 49 the extremities." in 1.1.
5.Please adjust Table 1 so that the duplicate Affected Skeletal Region can be combined into a text box and Different Radiation Patterns can be described separately.
6.In 2.2, the author proposed that "even if the first procedure result in a non-diagnostic result, a repetition of CT-guided biopsy is suggested before any other biopsy technique. "What is the role of repeated CT-guided biopsy? If in order to prevent miscalculation, does it mean that the accuracy of CT-guided biopsy is not high enough?
7.In 2.5.1, the author proposed that “To the best of our knowledge there are no series/original article focused on CT-guided biopsy of the hand and foot. Anyway, the procedure is known to be feasible also in these skeletal regions.” Please give a reason why it is feasible. If it is feasible, why has it not been adopted yet?
8.Chapter 2 is merely a summary of cases of CT-guided bone biopsies used in multiple human sites. It is suggested to add a short summary at the end of each section/chapter.
9.The content summarized in the chart lacks the supporting evidence of the literature. For example, Table 1, Figure 2, and table 2.
10.The conclusion is a little sloppy. In addition to summarizing the content of this paper, please add prospects or more detailed suggestions for clinicians.
Author Response
Thank You for the overall appreciation of our manuscript; here are our point-by-point responses:
- Thanks. Corrected.
- We have now corrected all the chapter and subchapter numbering and titles.
- We declare that all the figures showed are of our proprieties, obtained in our Institution, original and never published. We have now adjusted all the figures format.
- References / citations have been added to Introduction section.
- Table 1 has been adjusted. No duplication titles.
- With the repetition of a biopsy a diagnostic result can be achieved at the second procedures only. Anyway, the authors we cited already considered this in calculation of diagnostic accuracy, as usually is performed in this biopsy research articles.
- Thank You for your correct suggestion. Our statement is based on articles of miscellaneous biopsies including feet. We have now specify it according to your suggestion.
- Yes, this section is a mere report of results in different skeletal sites, and the conclusions and comments are in the following paper's sections. This is how the paper was organized.
- We implemented references and biobliographic articles, and now we feel that our figures, tables, and chart are well supported by the literature evidences.
- Thank You, according to your suggestion we implemented and improve the conclusion section of the manuscript.